# Structural insights into human topoisomerase 3β DNA and RNA catalysis and nucleic acid gate dynamics

Xi Yang [1], Xuemin Chen [2,3], Wei Yang [2] ✉ & Yves Pommier [1] ✉

Type IA topoisomerases (TopoIAs) are present in all living organisms. They resolve DNA/RNA catenanes, knots and supercoils by breaking and rejoining single-stranded DNA/RNA segments and allowing the passage of another nucleic acid segment through the break. Topoisomerase III-β (TOP3B), the only RNA topoisomerase in metazoans, promotes R-loop disassembly and translation of mRNAs. Defects in TOP3B lead to severe neurological diseases. We present a series of cryo-EM structures of human TOP3B with its cofactor TDRD3 during cleavage and rejoining of DNA or RNA, thus elucidating the roles of divalent metal ions and key enzyme residues in each step of the catalytic cycle. We also obtained the structure of an open-gate configuration that addresses the long-standing question of the strand-passage mechanism. Our studies reveal how TOP3B catalyzes both DNA and RNA relaxation, while TOP3A acts only on DNA.

Type IA topoisomerases (TopoIAs) are present across all living organisms[1–4], regulating DNA and RNA topology in replication, transcription, recombination, and translation[2,3,5,6]. They belong to three subfamilies: TOP1 (in bacteria), TOP3 (in bacteria, archaea and eucharia) and Reverse Gyrase (in archaea and hyperthermophilic eubacteria)[2,7–9]. Their core consists of four conserved domains arranged in a basket shape[1,2]. Domains I, III and IV form the base of the basket with a single-strand nucleic acid binding groove. The catalytic center straddles domain III containing the catalytic tyrosine, and domain I containing catalytic metal-ions[2,10]. Domain II forms the handle that bridges domains III and IV, creating a central cavity (Fig. 1e). Unlike the core, the C-terminal domains vary significantly among TopoIAs[5,11].

Without ATPase activity, TOP3s and bacterial TOP1s can relax negatively supercoiled DNA and resolve entangled DNA and RNA in a single-stranded region[12,13] by cleaving and rejoining one DNA/RNA strand and allowing passage of another strand[3,14–18]. The strand passage mechanism requires opening of the topo-gate between domain III and domains I and IV mediated by a hypothetical hinge in domain II[19–22] (Fig. 1e). Cleavage and end-rejoining are general acid-base reactions. The nucleophilic tyrosine attacks the scissile phosphate to cleave

ssDNA/RNA while covalently binding to the 5′-end of the DNA/RNA, leaving a 3′-OH group at the other end, which acts as the nucleophile upon end-rejoining[15]. Various numbers of divalent cations are required for several TopoIA enzymes in breaking and rejoining DNA[10,23,24], but their precise positions and functions have remained elusive[25,26]. Additionally, fundamental questions about type IA topoisomerases persist, including: 1) how they cleave and rejoin RNA; 2) how the topo-gate opens to initiate strand passage; and 3) the roles of their C-terminal domains.

Here we studied human TOP3B, which has critical functions in regulating cellular R-loops and mRNA metabolism[5,6,27,28], ensuring genome stability, neurodevelopment, and normal aging[29,30]. Its multifaceted activities are associated with a scaffold protein, TDRD3, that guides TOP3B to its cellular targets[27,28,30,31] through direct interactions with domain II of TOP3B[22] (Fig. 1a, e).

We acquired cryo-EM structures of TOP3B in various DNA and RNA catalytic states (Table 1) and characterized its DNA and RNA catalytic cycles. We captured an open-gate conformation providing insights into the strand-passage mechanism required for DNA/RNA relaxation and decatenation by TopoIA. Our TOP3B-TDRD3 structures

[1]Developmental Therapeutics Branch & Laboratory of Molecular Pharmacology, Center for Cancer Research, National Cancer Institute, NIH, Bethesda, MD, USA. [2]Laboratory of Molecular Biology, National Institute of Diabetes and Digestive and Kidney Diseases, NIH, Bethesda, MD, USA. [3]School of Life Sciences, Anhui University, Hefei, China. ✉e-mail: weiy@niddk.nih.gov; pommier@nih.gov

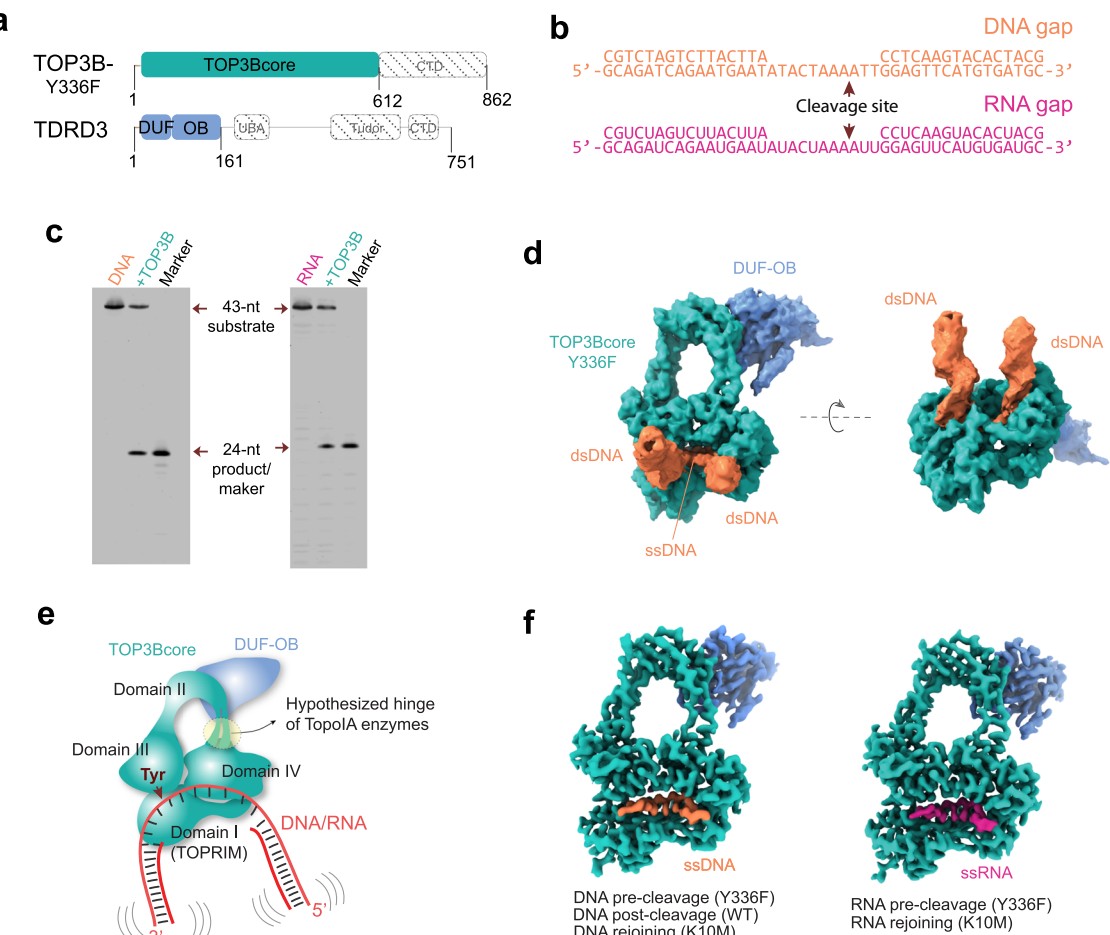

**Fig. 1 | Cryo-EM structures of cTOP3B complexing with DNA and RNA. a** Domain structures of TOP3B and TDRD3. Colored regions highlight the protein segments of the core complex of TOP3B-TDRD3 (cTOP3B) resolved from the cryo-EM volumes in (**d**) and (**f**). **b** 43-mer gapped DNA and RNA substrates including a 11-nt single-stranded region for cTOP3 binding. Arrows indicate strong TOP3B cleavage sites. **c** TOP3B cleavage assay with the gapped DNA and RNA substrates described in (**b**). 5′ ends of the cleavage strands were Cy3-labeled. The 43-nt and 24-nt ssDNA/RNA labels indicate substrates and cleavage products/markers on a representative urea-PAGE gel. The results shown in panel c are representative results obtained from two

independent experimental replicates. Source data are provided as a Source Data file. **d** Cryo-EM volume of cTOP3B-Y336F bound to gapped DNA (after applying a gaussian filter) in orthogonal views. The gapped RNA complex (not shown) resembles the DNA complex. **e** Schematic diagram depicting the cTOP3B-DNA/RNA complex as shown in (**d**). Red arrow indicates the catalytic tyrosine and cleavage site on the substrate. **f** High-resolution maps showing the cTOP3B-Y336F protein and ssDNA and ssRNA portions of the gapped substrates, which are representative of multiple cryo-EM resolved TOP3B intermediates (listed below) using wild-type and mutant cTOP3B.

also unveiled novel interactions and roles of TOP3B's C-terminal domain in DNA relaxation.

## Results

### Structures of TOP3B intermediates within DNA and RNA catalytic cycles

To elucidate TOP3B's DNA and RNA catalytic cycles, we set out to capture reaction intermediate states upon cleavage and rejoining using WT and mutant TOP3B. We co-expressed TOP3Bcore (1-612 aa) and the N-terminal fragment of TDRD3 (1-171 aa) containing its DUF domain and OB-fold (Fig. 1a) in HEK293 cells, purified them as a heterodimer (hereinafter referred to as cTOP3B), as TOP3B is more stable when bound to TDRD3.

We designed a 43-mer gaped substrate comprising an 11-nt single-stranded segment flanked by 16-bp duplexes (Fig. 1b). The single-stranded segment contains a TOP3B cleavage site[10] for both DNA and RNA (Fig. 1b, c). This allowed us to generate a TOP3B complex with RNA and compare DNA and RNA complexes. To obtain DNA- and RNA-bound TOP3B prior to cleavage, we substituted the nucleophile Y336, which forms tyrosyl-phosphate covalent bond with cleaved DNA, with

a phenylalanine (F), to disable nucleic acid cleavage while preserving the DNA/RNA interactions. We prepared the cryo-EM sample by mixing cTOP3B-Y336F with DNA/RNA gap substrates in equimolar concentrations, and resolved the DNA and RNA complexes at an overall ~3.3 Å resolution (Fig. 1d, e) revealing rigid components including protein and single-stranded nucleic acid fragments (Fig. 1f). Using the same strategy, we obtained a DNA post-cleavage complex with wild-type cTOP3B, and generated DNA and RNA rejoining complexes (Fig. 1f) with the rejoining-deficient mutant cTOP3B-K10M[10] (detailed in the corresponding section). The DNA/RNA ends are closely aligned in the rejoining complexes and more relaxed in the post-cleavage complex.

### TOP3B cleavage: one-metal-ion catalysis facilitated by an additional structural divalent cation

The roles of divalent metal ions, typically $Mg^{2+}$ ion, in TopoIA catalysis are not yet fully understood, and the necessity of a divalent cation for DNA cleavage has been long debated[2,10,26,32,33]. This is largely due to the inconsistent DNA cleavage activity of different TopoIA enzymes in the absence of divalent ions[10,15,26] and non-physiological conditions of

**Table 1 | Cryo-EM data collection, refinement, and validation statistics**

| | cTOP3B DNA pre-cleavage | cTOP3B DNA post-cleavage | cTOP3B DNA rejoining | cTOP3B DNA pre-cleavage | cTOP3B DNA post-cleavage | cTOP3B DNA rejoining | cTOP3B RNA pre-cleavage | cTOP3B RNA rejoining | cTOP3B dimer with DNA bubble | Full-length TOP3B-TDRD3 |
|---|---|---|---|---|---|---|---|---|---|---|
| | (DNA Gap 1 substrate) | | | (DNA Gap 2 substrate) | | | (RNA Gap substrate) | | | |
| PDB code | 9C9Y | 9CA0 | 9CA1 | 9CAJ | 9CAL | 9CAK | 9CAG | 9CA4 | 9C9W | 9CAH |
| EMDB code | EMD-45376 | EMD-45378 | EMD-45379 | EMD-45393 | EMD-45395 | EMD-45394 | EMD-45390 | EMD-45380 | EMD-45374 | EMD-45391 |
| **Data collection and processing** | | | | | | | | | | |
| Magnification | 100,000 | 105,000 | 100,000 | 100,000 | 100,000 | 105,000 | 105,000 | 100,000 | 100,000 | 130,000 |
| Pixel size (Å) | 0.83 | 0.83 | 0.83 | 0.83 | 0.83 | 0.83 | 0.83 | 0.83 | 0.83 | 0.83 |
| Defocus range (μm) | −0.8 to −2.4 | −0.8 to −2.4 | −0.8 to −2.4 | −0.8 to −2.4 | −0.8 to −2.4 | −0.8 to −2.4 | −0.8 to −2.4 | −0.8 to −2.4 | −0.8 to −2.4 | −0.8 to −2.4 |
| Voltage (kV) | 200 | 300 | 200.00 | 200 | 200 | 300 | 300 | 200 | 200 | 300 |
| Total electron dose (e⁻/Å²) | 46 | 54.4 | 56.00 | 56 | 56 | 54.4 | 54.4 | 46 | 46 | 50 |
| Symmetry imposed | C1 | C1 | C1 | C1 | C1 | C1 | C1 | C1 | C1 | C1 |
| Final particles in reconstruction (no.) | 720,245 | 198,867 | 901,517 | 452,048 | 996,542 | 770,197 | 140,000 | 432,855 | 63,358 | 469,366 |
| Map resolution (Å, FSC = 0.143) | 3.35 | 3.48 | 3.26 | 3.51 | 3.15 | 3.01 | 3.33 | 3.41 | 4.25 | 3.16 |
| **Model refinement and validation** | | | | | | | | | | |
| Initial model used (PDB code) | 5gve | 5gve | 5gve | 5gve | 5gve | 5gve | 5gve | 5gve | 5gve | 5gve |
| Mask CC | 0.79 | 0.78 | 0.80 | 0.76 | 0.79 | 0.79 | 0.79 | 0.78 | 0.69 | 0.78 |
| Volume CC | 0.8 | 0.78 | 0.80 | 0.77 | 0.8 | 0.8 | 0.79 | 0.79 | 0.69 | 0.78 |
| Peak CC | 0.74 | 0.73 | 0.74 | 0.74 | 0.74 | 0.76 | 0.75 | 0.73 | 0.66 | 0.75 |
| Model resolution (Å, FSC = 0.5) | 3.5 | 3.6 | 3.5 | 3.6 | 3.3 | 3.2 | 3.4 | 3.6 | 4.6 | 3.5 |
| **B-factors (Å)** | | | | | | | | | | |
| Protein | 85.39 | 75.29 | 82.73 | 63.68 | 74.5 | 59.81 | 71.35 | 86.22 | 134.86 | 72.64 |
| Nucleotide | 74.68 | 66.35 | 82.31 | 68.38 | 72.26 | 60.64 | 66.12 | 79.37 | 248.85 | N/A |
| Ligand | 82.64 | 67.96 | 76.81 | 78.85 | 83.77 | 64.59 | 70.1 | 75.39 | 165.83 | 86.7 |
| **R.m.s. deviations** | | | | | | | | | | |
| Bond lengths (Å) | 0.005 | 0.003 | 0.003 | 0.004 | 0.003 | 0.004 | 0.004 | 0.004 | 0.005 | 0.003 |
| Bond angles (°) | 0.663 | 0.537 | 0.547 | 0.663 | 0.529 | 0.567 | 0.627 | 0.627 | 1.050 | 0.613 |
| **Ramachandran plot** | | | | | | | | | | |
| Favored (%) | 93.89 | 95.82 | 95.69 | 95.19 | 96.08 | 95.17 | 95.19 | 94.65 | 90.23 | 92.58 |
| Allowed (%) | 5.98 | 4.18 | 4.18 | 4.81 | 3.92 | 4.83 | 4.68 | 5.22 | 9.58 | 7.20 |
| Outliers (%) | 0.13 | 0.00 | 0.13 | 0.00 | 0.00 | 0.00 | 0.13 | 0.13 | 0.20 | 0.22 |
| **Validation** | | | | | | | | | | |
| Clash score | 6.71 | 6.67 | 5.13 | 12.49 | 5.47 | 7.48 | 9.53 | 6.71 | 18.78 | 8.39 |
| Molprobity score | 1.78 | 1.66 | 1.57 | 1.94 | 1.56 | 1.75 | 1.84 | 1.74 | 2.31 | 1.92 |
| Rotamer outliers (%) | 0.15 | 0.15 | 0.00 | 0.45 | 0.00 | 0.00 | 0.15 | 0.00 | 0.15 | 0.13 |
| C-beta outliers (%) | 0.00 | 0.00 | 0.00 | 0.00 | 0.00 | 0.00 | 0.00 | 0.14 | 0.00 | 0.00 |

crystallization and lattice contacts of the X-ray diffraction methods[26,33,34].

Mg²⁺ and Mn²⁺ both activate TOP3B in nucleic acid cleavage and rejoining, with Mn²⁺ being more effective[10] owing to its less stringent coordination requirement and better ability to stabilize certain reaction intermediates. Additionally, the higher electron count of Mn²⁺ can enhance visibility in cryo-EM maps, making it the preferred choice for our study. Our cryo-EM structure of the DNA pre-cleavage complex of TOP3B revealed two Mn²⁺ ions (Fig. 2a, b). One Mn²⁺ ion (Mn$_C^{2+}$; "c" stands for catalysis) is coordinated by four amino acid residues and the DNA scissile phosphate (Fig. 2a). A divalent cation at the Mn$_C^{2+}$ site was

proposed to play a catalytic role in DNA/RNA cleavage[10] and to be captured before binding nucleic acids[21,22]. We found it slightly repositioned upon DNA binding (Supplementary Fig. 1a). The active-center geometry of TOP3B is reminiscent of nucleases and recombinases that employ a one-metal-ion mechanism to break nucleic acid backbones[35], such as the HUH endonuclease TraI[35,36]. These enzymes use a single divalent cation located near the 3′-O leaving-group to facilitate the phosphoryl transfer. Mn$_C^{2+}$ also resembles metal ion B in the classical two-metal-ion catalysis enzymes[35,37].

The second active-site divalent ion (Mn$_S^{2+}$; "s" stands for structure) is coordinated by a different set of residues and is ~4.7 Å away

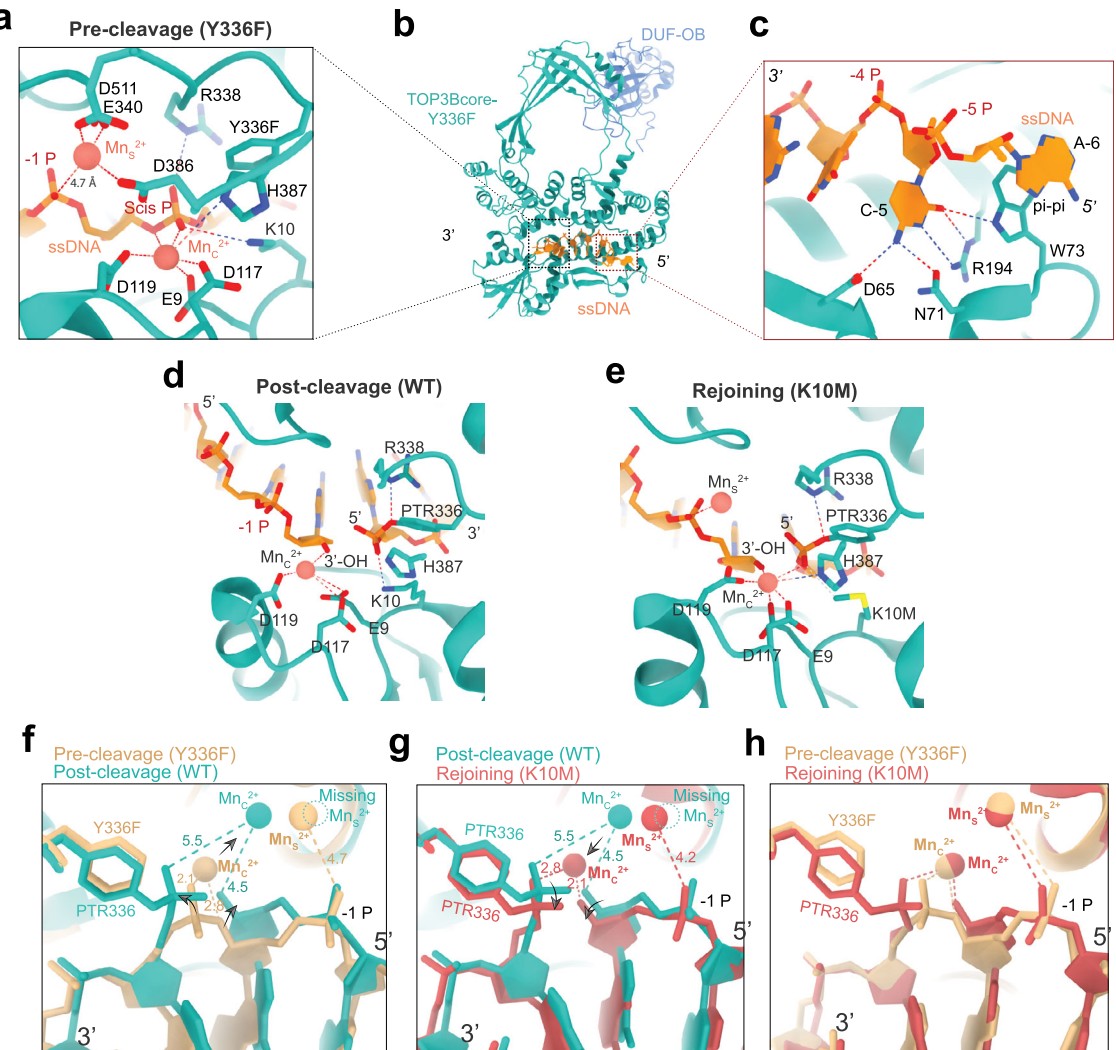

**Fig. 2 | Active-site configurations of TOP3B within a DNA catalytic cycle.**
**a** Active-site divalent-cation coordination in the pre-cleavage state. $Mn_C^{2+}$: the catalytic cation, coordinated by E9, D117, D119, H387 and a non-bridging oxygen and a bridging oxygen (leaving group) in the DNA scissile phosphate (Scis P). $Mn_S^{2+}$: the structural cation, coordinated by E340, D386, D511, and is ~4.7 Å away from the −1 phosphate group (−1 P). Y336F: the phenylalanine substitution for the catalytic tyrosine of the wild-type (WT) enzyme. Doted lines indicate molecular interactions. **b** Pre-cleavage complex (Y336F mutant). **c** Interactions of TOP3B with the cytosine at position −5 (C-5) and adenine −6 (A−6). Dotted lines represent salt bridges. The aromatic ring of W73 π−π stacks with A-6. **d**, **e** Post-cleavage and rejoining

configurations of cTOP3 and the K10M mutant. PTR336: the tyrosyl-phosphodiester linkage. **f** Superimposition illustrating structural changes before (yellow) and after (green) DNA cleavage. Arrows indicate separation of the cleaved DNA ends and displacement of $Mn_C^{2+}$ after cleavage. Numbers denote the measured distances (Å). Dotted circle highlights the missing $Mn_S^{2+}$. **g** Superimposition of the DNA post-cleavage (green) and rejoining (red) complexes showing convergence (arrows) of the cleaved DNA ends and $Mn_C^{2+}$. **h** Superimposition of the DNA rejoining and pre-cleavage (equivalent to post-rejoining) complexes indicates an SN2-type phosphoryl transfer that joins DNA.

from the -1 phosphate (Fig. 2a), potentially interacting with the phosphate group through water molecules in its hydration shells. Therefore, $Mn_S^{2+}$ appears to play a structural role in positioning the -1 phosphate, facilitating the alignment of the nearby scissile phosphate for DNA cleavage. The protein side chains coordinating the second divalent ion are conserved in human TOP3A and *E. coli* TOP3 (Supplementary Fig. 1b) but absent in *E.coli* TOP1, suggesting distinct mechanisms between TOP3 and bacterial TOP1.

Despite the conservation of the $Mn_C^{2+}$ binding site in both TOP3 and bacterial TOP1 (Supplementary Fig. 1c), the crystal structure of *E.coli* TOP3-ssDNA lacks the catalytic metal ion[33]. This is possibly due to the low pH condition (pH 5.5) of crystallization that caused protonation of the carboxylates of the metal ion-coordinating residues as well as the scissile phosphate. The active-site geometry of the low-pH and protonated *E.coli* TOP3 structure closely resembles that of TOP3B (Supplementary Fig. 1d). The active site of bacterial TOP1 is nearly

superimposable with that of TOP3B, as the $Mg^{2+}$-bound catalytic site of MtbTOP1 (Supplementary Fig. 1e). In addition, the two-metal ion mechanism of TOP3B resembles that of Yeast TOP2[32].

Our structures reveal that the Y336F points towards the scissile phosphate, reflecting the geometry of the catalytic tyrosine (Fig. 2a) while R338 and K10 interact with the non-bridging oxygen atoms of the scissile phosphate, positioning and polarizing the scissile phosphate for the nucleophilic attack and subsequently stabilizing the transition state pentavalent phosphate (Supplementary Fig. 2a, b). The arginine is conserved in all TopoIAs, while a lysine at this specific position within reach of the DNA backbone is exclusive to the TOP3 subfamily[26,33]. Accordingly, mutating the arginine in bacterial TOP1 or the lysine in human TOP3B results in increased divalent-metal-ion dependence for DNA cleavage[10,38]. These mutations alter the alignment and charge environment of the scissile phosphate. Without the lysine residue, bacterial TOP1[26] may have a substitution such as a monovalent ion to

facilitate the active-site alignment. Indeed, a potassium ion in RNase H1 fulfills the role of a lysine residue in a comparable catalytic site of mammalian Endonuclease V[39].

## Molecular basis for TOP3B sequence selectivity

TOP3B prefers certain DNA sequences for binding and cleavage. Strong cleavage sites contain a cytosine at the -5 position (C-5)[10]. Our structure of cTOP3B-DNA reveals several interactions between C-5 and surrounding amino acid residues through salt bridges and hydrogen bonds (Fig. 2c). We also observed π−π stacking between base -6 and the W73 aromatic ring, consistent with a strong hydrophilic interaction with one base, combined with π−π stacking with the adjacent base observed in other Type IA topoisomerases[15,26,33], including nucleobases -5 and -6 for TOP3 and -4 and -5 for bacterial TOP1[26,40]. Stable binding may ensure a secure attachment to the non-covalently bound DNA/RNA-end after cleavage and prevent irreversible DNA/RNA cleavage. Yet, despite a strong preference for the C-5 base, TOP3B retains the flexibility to accommodate varied DNA bases surrounding C-5, as evidenced by the comparison of two structures of cTOP3B with different DNA substrates (Supplementary Fig. 4b,c).

## Conformational transitions of TOP3B before and after DNA cleavage

Divalent metal ions accelerate TOP3B in nucleic acid binding, cleavage and rejoining[10]. At relatively low divalent-cation concentration (e.g. <1 mM), the cleaved DNA or RNA intermediates dominate while increasing the divalent-cation concentration favors the rejoining reaction (Supplementary Fig. 1f, g).

The DNA cleaved state was visualized by cryo-EM of cTOP3B-DNA in 0.5 mM $Mn^{2+}$ (Fig. 2d, f) with the catalytic Y336 covalently linked to the DNA scissile phosphate. This post-cleavage TOP3B state exhibits a relatively relaxed conformation with separated DNA ends and displaced $Mn_C^{2+}$ (Fig. 2f). DNA cleavage reshapes the geometry of the coordination between DNA and $Mn_C^{2+}$ with $Mn_C^{2+}$ shifting away from the non-bridging oxygen (from 2.1 Å to 5.5 Å) and from the leaving group/3'-OH end (from 2.8 Å to 4.5 Å). Notably, the $Mn_S^{2+}$ disappears after DNA cleavage (Fig. 2f), suggesting that protein conformational changes after cleavage destabilize $Mn_S^{2+}$ binding.

## DNA end-rejoining is facilitated by $Mn_S^{2+}$ re-capture

A DNA/RNA end-rejoining state of a Type IA topoisomerase had not been visualized. To obtain a DNA rejoining complex of TOP3B, we employed the rejoining-deficient cTOP3B-K10M[10], which cleaves DNA and RNA slowly and is unable to rejoin DNA (Supplementary Fig. 1h, i), likely due to lack of K10-assisted phosphoryl transfer. We anticipated that exposing cTOP3B-K10M to elevated divalent cation concentrations, known to stimulate DNA and RNA rejoining, would arrest the mutant in a rejoining conformation. Accordingly, the cryo-EM structure of cTOP3B-K10M as DNA rejoining intermediate (Fig. 2e, g) showed the cleaved DNA ends closer to each other than in the post-cleavage complex (Fig. 2g) with the 3'-OH group 3.1 Å from the tyrosyl-phosphate poised for the reverse nucleophilic attack. Compared with the post-cleavage complex, the catalytic $Mn_C^{2+}$ shifts towards both the scissile phosphate (from 5.5 Å to 2.8 Å) and the 3'-OH (from 4.5 Å to 2.1 Å) and bridges these two reactive groups. Notably, the second cation $Mn_S^{2+}$ is re-captured in this state.

To rejoin the cleaved DNA, the 3'-OH-end must act as the nucleophile attacking the phosphotyrosyl bond. At this step, we observed (Fig. 2e and Supplementary Fig. 2d) that $Mn_C^{2+}$ aligns the 3'-OH group with the scissile phosphate and facilitates the hydroxyl group deprotonation. Thus, $Mn_C^{2+}$ resembles the A-site divalent cation in DNA/RNA polymerases[37,41,42]. However, unlike polymerases, TOP3B lacks a firmly bound B-site metal ion on the leaving group side. Instead, it utilizes both the positively charged R338 and K10 residues to assist the phosphoryl transfer (Supplementary Fig. 2d, e), consistent with the

fact that mutating either R338 or K10 leads to impaired DNA re-ligation[10,25]. Superposition of the rejoining complex and the pre-cleavage complex (equivalent to post-rejoining complex) reveals the transitions from the tyrosyl-phosphate bond to the DNA phospho-diester bond in accordance with SN2 reaction stereo-chemistry (Fig. 2h).

The re-capture of $Mn_S^{2+}$ for DNA rejoining signifies its indispensable role in coordinating D386 and E340 in domain III, D511 in domain IV, along with the DNA -1 phosphate (Supplementary Fig. 3a, b). The binding of $Mn_S^{2+}$ stabilizes a compact enzyme conformation where domain III, IV and I are close to each other, compared with the relatively relaxed conformation in the post-cleavage state (Supplementary Fig. 3b and Fig. 2g). Both conformations likely dynamically interconvert, with the compact conformation favoring $Mn_S^{2+}$ association. Additionally, this $Mn_S^{2+}$-binding-induced conformational change is coupled with the juxtaposition of the DNA ends and repositioning of the catalytic cation $Mn_C^{2+}$ for DNA rejoining (Fig. 2g and Supplementary Fig. 3c). This is in line with our observation that high $Mn^{2+}$ concentration induces rapid DNA/RNA rejoining by TOP3B (Supplementary Fig. 1f,g). Again, the absence of $Mn_S^{2+}$ metal ligands in bacterial TOP1 (Supplementary Fig. 1b) suggests mechanistic differences between TOP1 and TOP3 rejoining.

## RNA binding and processing by TOP3B

The RNA topoisomerase activity of TopoIA enzymes has been identified across all three domains of life[5], with the exception of human TOP3A, some Mycobacterial TOP1s and reverse gyrases[5,43]. TOP3B-TDRD3 with FMRP have been found to associate with polyribosomes in metazoans[43], and to regulate translation for neurodevelopment[27,28]. Yet, RNA complexes of TopoIA have not been previously visualized. We obtained RNA pre-cleavage and rejoining complexes of TOP3B. As for ssDNA, TOP3B interacts with the ssRNA backbone through its charged amino-acid side chains, divalent-cation cofactors, and alpha-helix dipole moments. In addition, both DNA and RNA form multiple intra-strand base stacking, facilitating backbone alignment (Fig. 3a, b). The backbone and base geometries of ssRNA and ssDNA are similar, with ssRNA establishing more backbone interactions. This agrees with our earlier observation that RNA binding/cleavage by TOP3B is less reliant on specific sequences than DNA[10].

RNA is A-form and features a shorter base-to-base distances compared to the B-form DNA due to its differential sugar pucker conformation and 2'-OH[44]. We observed that in the ssRNA bound to TOP3B structure, the upstream and downstream phosphate groups and bases are closer to the central scissile phosphate than those in ssDNA (Fig. 3d). Consequently, TOP3B exhibits a more compact structure with ssRNA (Fig. 3c and Supplementary Video 1) than ssDNA, suggesting conformational flexibility upon binding to RNA or DNA.

We hypothesized that the differential flexibility within the TopoIA family may explain why TOP3A and some MycoTOP1 enzymes cannot process RNA[43]. The likely tighter hydrophobic core of these enzymes may restrict their ability to contract sufficiently to bind ssRNA. To test this possibility, we examined TOP3A cleavage using a 15-nt ssDNA substrate containing a shared cleavage site with TOP3B (Supplementary Fig. 2). Replacing a single deoxyribonucleotide with a ribonucleotide at the cleavage site did not significantly affect the cleavage and rejoining activities of TOP3Acore (Supplementary Fig. 2h, i), demonstrating that the active-site chemistry of TOP3A is not perturbed by switching to an RNA substrate. Additionally, we found that TOP3A can accommodate four ribonucleotides around the cleavage site, but not six (Supplementary Fig. 2j, k), while TOP3B readily cleaves both substrates. These results reinforce our hypothesis that TOP3A has limited conformational flexibility, and it cannot accommodate the compaction required for RNA binding.

The catalytic sites of TOP3B are superimposable in the pre-cleavage DNA and RNA complexes (Fig. 3e), indicating a common

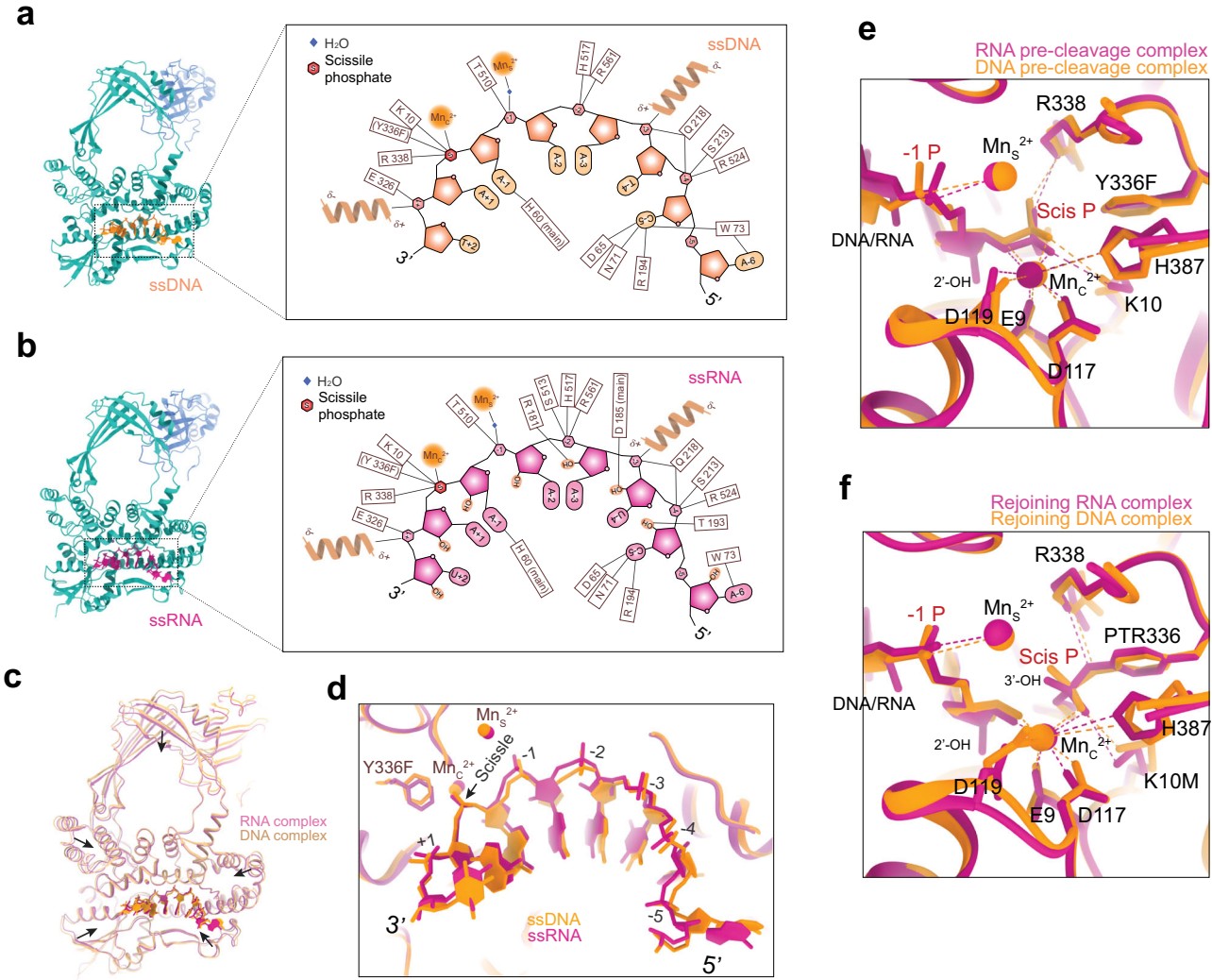

**Fig. 3 | Comparison of RNA and DNA binding, cleavage, and rejoining.**
**a**, **b** Schematic depiction of the interactions of ssDNA and ssRNA with TOP3B. Helical coils represent α-helix dipole moments interacting with the corresponding phosphate groups. "(main)" indicates interactions via main-chain groups. Numbers label positions of phosphate groups and bases. **c** Superimposition of DNA and RNA complexes of cTOP3-Y336F showing contracted RNA-bound conformation (arrows) compared with DNA-bound conformation (also shown in Supplementary Video 1). **d** Expanded view of (**c**) showing ssDNA and ssRNA within TOP3B complexes. Numbers indicate positions of DNA/RNA phosphate groups. **e**, **f** Active-site configuration alignments of the DNA and RNA pre-cleavage and rejoining states of TOP3B. $Mn_S^{2+}$ is present in both the DNA and RNA complexes.

mechanism employed by TOP3B for cleaving DNA and RNA (Supplementary Fig. 2a, f). Yet, because we recently reported that the DNA-rejoining-deficient K10M mutant can still slowly rejoin RNA[10], we aimed to explain its enhanced RNA rejoining activity by determining the structure of an RNA rejoining complex with cTOP3B-K10M. We observed a highly similar active-site conformation for both (Fig. 3f), implying a shared mechanism of TOP3B for DNA and RNA rejoining (Supplementary Fig. 2d, g). Therefore, the enhanced RNA rejoining activity may relate to the ribose 2'-OH group adjacent to the nucleophile 3'-OH making it more susceptible to deprotonation. In addition, the presence of $Mn_S^{2+}$ ion in both the DNA and RNA rejoining complexes underscoring its crucial role in repositioning the protein domains and nucleic acid ends for rejoining.

## Structure of an open-gate TOP3B
The gate-opening mechanism of Type IA topoisomerases have been investigated using diverse approaches. A 30-kD fragment of *E. coli* TOP1 containing domains II and III was crystallized in various conformations with two hypothetical enzyme hinge regions[20], and a recent magnetic tweezers single-molecule study reported the separation of the cleaved DNA ends in *E. coli* TOP1 and TOP3[45]. However, the structure of an open-gate conformation of a complete TopoIA core has not yet been visualized.

To capture the strand passage event or the passing DNA in the large central cavity of the enzyme, we replaced the gapped 43-mer DNA substrate with a DNA bubble, in which the 11-nt ssDNA is opposite to 13 unmatched nucleotides. Mixing the DNA substrate with cTOP3B at an approximately 1:1 ratio showed a primary particle population with a single cTOP3B forming a cleavage complex on the 11-nt strand resembling the monomeric complex generated with the DNA gap. Additionally, we identified species with two cTOP3B bound to one DNA bubble and obtained a ~4.3 Å cryo-EM density map of this cTOP3B dimer from 63,000 particles (Fig. 4a). Within this complex, one enzyme binds to the 11-nt strand and forms a cleavage complex. The second enzyme binds to the opposite 13-nt strand without cleaving DNA (Fig. 4b, Supplementary Fig. 5a and Supplementary Video 2). Remarkably, the second cTOP3B assumes an open-gate conformation with a dsDNA arm trapped in the gap between domains I and III and interacting with domains III and II via charged residues (Supplementary Fig. 5c). Interactions between domains III of two TOP3B molecules

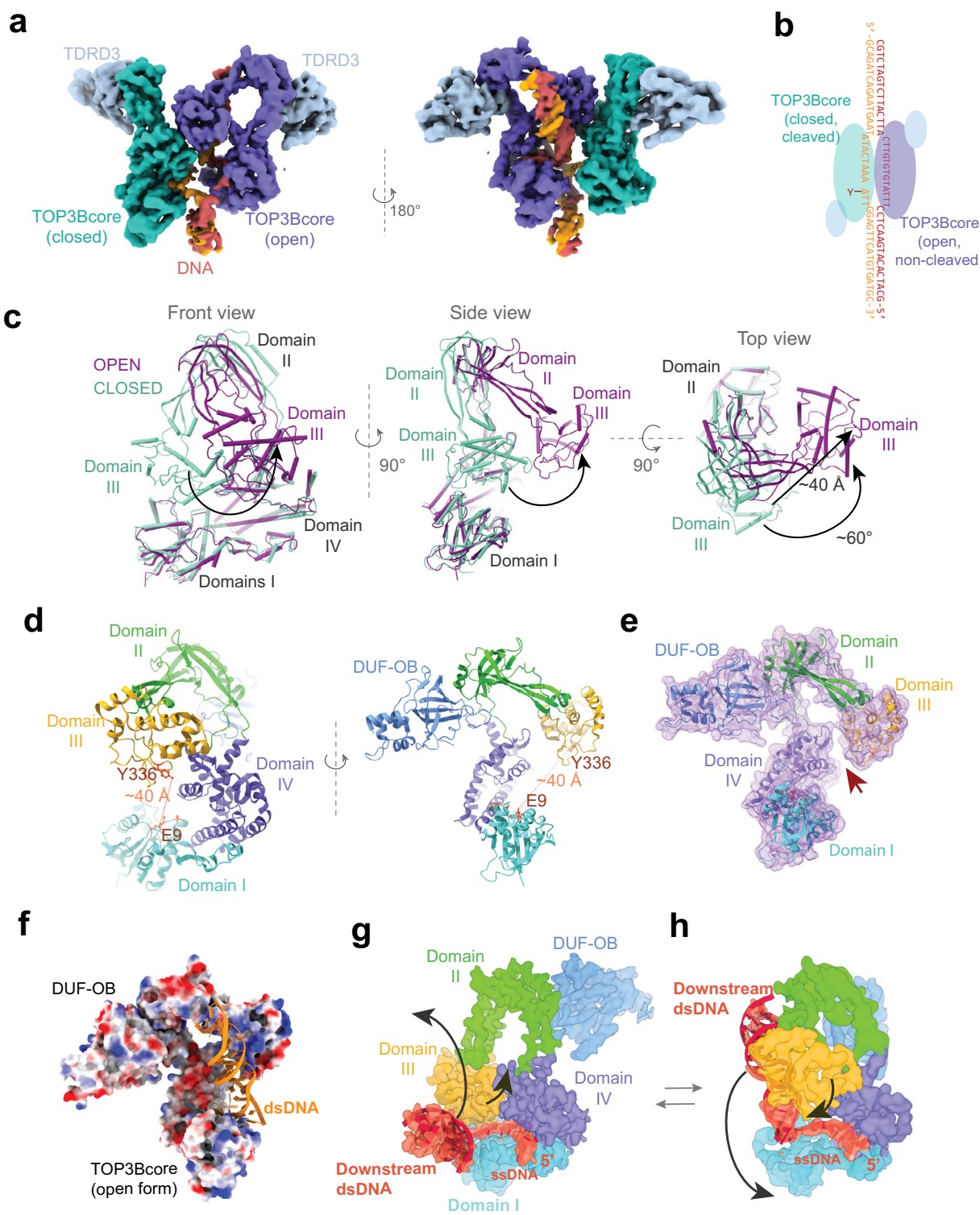

provide additional stability to this open-gate conformation (Supplementary Fig. 5b).

We next compared the structure of the open-gate cTOP3B from the dimer complex with the closed cTOP3B conformation in the DNA pre-cleavage complex (Fig. 4c). While the closed cTOP3B exhibits a planar ring conformation, switching to the open conformation induces a major swing of domains II and III with rotation of domain III over 60° coupled with a ~40 Å translation that opens the planar ring (Fig. 4c,

Supplementary Videos 3 and 4). The catalytic tyrosine in domain III disengages from the $Mn_C^{2+}$ binding site in domain I, creating a 40 Å gap between them (Fig. 4d), which may represent the open ssDNA gate of TOP3B in the DNA-cleaved state. This conformational transition also disassembles the hydrophobic core between domains III and IV, leaving only a weak charge interaction between them (Fig. 4e).

The out-of-plane opening of TOP3B is reminiscent of a ring-shaped sliding DNA clamp open by a clamp loader[46]. The former

**Fig. 4 | Transition states between the close and open conformations of TOP3B.**
**a**, **b** Cryo-EM map and schematics showing two cTOP3B binding to a mismatch bubble DNA construct including a DNA-cleavage and closed state cTOP3B (green) and an open state cTOP3B without DNA cleavage (purple). One side of the DNA bubble contains a strong TOP3B cleavage site cleaved by the green cTOP3B. The opposite strand lacks a TOP3B cleavage site. **c** Alignment of the TOP3Bcore open conformation (extracted from the complex shown in A) and of the closed TOP3Bcore (from a DNA pre-cleavage complex) reveals a swinging motion of domains III and II (arrows). Numbers indicate the measured domain III rotation and shift between the two conformations. **d** The open conformation of cTOP3B with colored domains. Dotted lines denote the distance (~40 Å) between the catalytic

Y336 and residue E9 coordinating $Mn_C^{2+}$. **e** Protein surface representation of the cTOP3B open conformation showing the partially opened domain III-IV gate (arrow). **f** Electrostatic surface potential of cTOP3B in the open conformation with a dsDNA arm trapped. The dsDNA interacts with the positively charged surfaces (blue) of domains II and III. **g**, **h** Conformational switch between the closed cTOP3B (complexed with a DNA gap) and the open cTOP3B. Map density and model in red highlight the ssDNA cleavage strand and downstream dsDNA within the DNA gapped/bubble substrate. Conformational changes of the dsDNA are coupled with the opening and closing of the topo gate (arrows). The downstream dsDNA within full-length TOP3B (closed state) complexed with a DNA bubble shows a similar orientation (Fig. 6b).

switches from a planar ring to a spiral conformation resembling the shape of a "lock washer". This mechanism can be an energetically favored way to open a ring-shaped protein assembly. It may preserve maximal interactions at the hinge in contrast to an in-plane ring opening that would disrupt those interactions. Actually, many hexametric helicases adopt similar structures in DNA loading and translocation[47,48].

Remarkably, we can also visualize the trend of dynamic topo-gate opening and closing within a catalytic cycle of TOP3B (Supplementary Fig. 6f–i), where domains III and II shift away from domain I after DNA cleavage and revert upon DNA rejoining.

### Hinge loops of TOP3B

Domains I and IV in the open and closed forms of cTOP3B are largely superimposable (Fig. 4c), as are domains II and III (Supplementary Fig. 6a). The hinge for gate opening is formed by two peptide loops connecting domains II and IV (Supplementary Fig. 6b): Loop 1 (residues 488–494) and Loop 2 (residues 235–238). These two loops are consistently observed in the TopoIA family and were previously hypothesized as a potential hinge[19–22]. Additionally, TDRD3's N-terminal DUF-OB-fold binds next to the hinge, and swings with TOP3B's domain III upon topo-gate opening without internal structural changes (Supplementary Figs. 6c, d and 5d). Thus, TDRD3 appears to stabilize the top region of domain II around the hinge and to support the gate opening and closing motion. The N-terminus of RMI1 and TDRD3 share a similar folding and interaction mode with TOP3A and TOP3B (Supplementary Fig. 6e), suggesting that RMI1 likely serves the same role as TDRD3 in supporting TOP3A's gate dynamics.

### Opening and closure of cTOP3B are coupled with the swinging motion of the downstream dsDNA

In our open TOP3B structure, the downstream dsDNA is trapped between domains I and III by electrostatic interactions with a highly positively charged surface of domains II and III (Fig. 4f and Supplementary Fig. 5c). This positive-charge surface is conserved in the TopoIA family (Supplementary Fig. 5e). We observed that when the gate is closed, the downstream dsDNA is often oriented nearly 60° perpendicular to the closed TOP3B planar ring (Supplementary Fig. 5f), and the adjacent ss/ds junction interacts with the enzyme by hydrogen bonds between domain I and DNA bases (Supplementary Fig. 5g). These open and closed conformations likely represent two intermediate states in the DNA relaxation cycle, when TOP3B is bound to a DNA bubble on negatively supercoiled DNA. The enzyme may transition between these conformations during DNA relaxation, while the dsDNA undergoes swinging movements (Fig. 4g, h). This DNA relaxation process also requires the upstream dsDNA to be constrained by the CTD of TOP3B as detailed below.

### The CTD of TOP3B stabilizes the TOP3B-TDRD3 complexed with DNA

The CTDs of type IA topoisomerases vary in length and sequence[2,5,11] and only a few have been structurally resolved[11]. The predicted CTD of

TOP3B encompasses multiple zinc-finger-like motifs (AlphaFold DB O95985)[49,50] and an arginine-rich tail[10], and the CTD of TOP3B has been shown to enhance the topoisomerase activity and interaction of TOP3B with TDRD3[10,22,30,31].

To study the structures of full-length TOP3B-TDRD3 in the presence and absence of DNA, we purified the human TOP3B(Y336F)-TDRD3 heterodimer from HEK293T cells. We first obtained an apo form TOP3B-TDRD3 map at ~3.1 Å resolution, which revealed the structure of TOP3B spanning amino acids 1–717, including half of its CTD (residues 612–717) as well as the N-terminal region of TDRD3 (residues 1–190) (Fig. 5a, b). The planar assembly of the protein complex exhibits substantial positively charged surface areas on one side (Supplementary Fig. 7), which may enhance its binding to nucleic acids. Three $Zn^{2+}$-binding motifs within the CTD of TOP3B were well defined (Fig. 5c, d), but the fourth and fifth C-4 zinc fingers (residues 718–815), as predicted by AlphaFold (AlphaFold DB O95985)[49,50], are too mobile to be captured.

Different from Zinc finger or zinc finger-like motifs in *E. coli*TOP1[11,51] and MtbTOP1[52], our structure reveals that TOP3B's CTD establishes intramolecular contacts with both the core (Fig. 5e) and an extended loop (residues 168–190) of TDRD3 (Fig. 5f). The TOP3B-TDRD3 heterodimer is further stabilized by TOP3Bcore-TDRD3 contacts involving hydrophobic interactions of the extended loop of TDRD3 with two antiparallel TOP3B α-helices of domain IV (Fig. 5g).

We resolved the full-length TOP3B(Y336F)-TDRD3(1-190 aa) in complex with a DNA mismatch bubble (two unpaired DNA strands), which can form in negatively supercoiled DNA[53,54]. The DNA bubble substrate was based on the 43-nt DNA gap (Fig. 1b) with addition of 11 mismatched nucleotides opposite to the cleavage strand (Fig. 6a). The resulting cryo-EM volume revealed binding of the TOP3Bcore to the cleavage strand in the mismatch bubble. Additional mapped volume adjacent to the TOP3Bcore showed interactions of the CTD with the upstream dsDNA (Fig. 6b, c). The structure reveals that, upon DNA binding, the CTD moves away from the TOP3Bcore and TDRD3 and binds to the DNA upstream of the cleavage site. We subsequently created a short R-Loop substrate with sufficient rigidity to elucidate the binding of the CTD to dsDNA near the TOP3Bcore (Fig. 6d–f). These results suggest that the CTD promotes the formation of a TOP3B-substrate complex, and that, in the context of DNA relaxation, binding and constraining the dsDNA upstream of the cleavage site can facilitate topological rearrangement of the downstream DNA (Fig. 7).

### A "sequential strand-passage" mechanism for TOP3B-mediated DNA relaxation

The gate opening-and-closing of TOP3B together with the swinging motion of the downstream dsDNA, suggests the following "sequential strand-passage" mechanism for DNA supercoil relaxation by TOP3B. TOP3B cleaves ssDNA (G-strand) within a locally unwound DNA region (negatively supercoiled), the cleaved 5′-end of DNA covalently attaches to domain III and the 3′-end tightly binds to domains I, IV, and the CTD (Fig. 7a), the 5′-end tends to rotate relative to the 3′-end, around the opposite strand (T-segment), to alleviate tortional stress, imposing

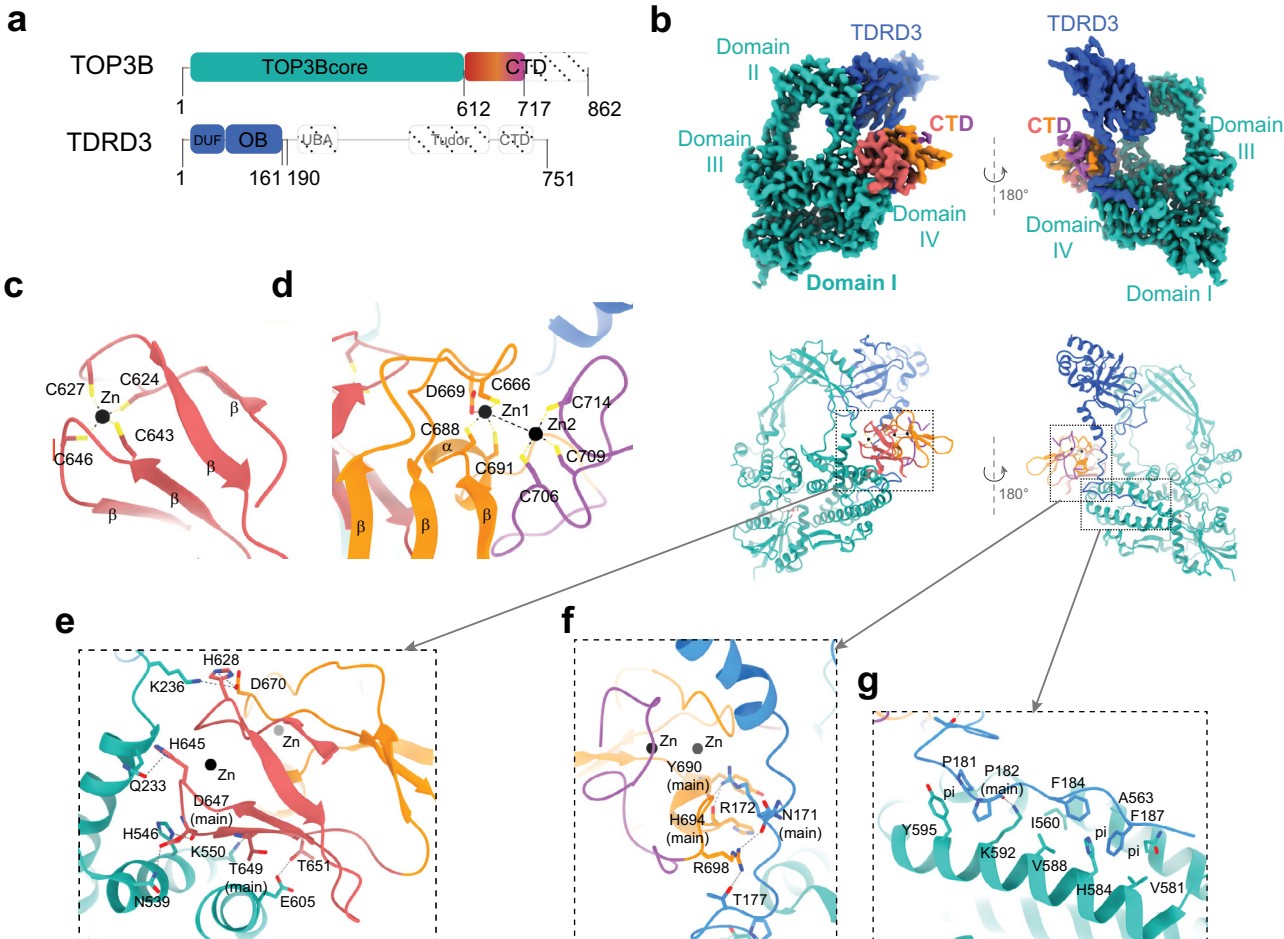

**Fig. 5 | Structure of apo form TOP3B-TDRD3. a** Colored regions indicate structurally resolved segments of TOP3B and TDRD3. **b** Cryo-EM map and structure of the coplanar TOP3B-TDRD3 complex. The majority of TDRD3 was not resolved, likely due to its inherent flexibility or disorder possibly due to the lack of additional protein partners. **c** C−4 type Zinc finger in the CTD of TOP3B containing four antiparallel β-sheets and a zinc ion coordinated by four cysteine residues. **d** Two additional Zinc binding motifs in the TOP3B CTD. One is comprised of three β-sheets and a short α-helix interconnected by peptide loops with the $Zn^{2+}$ (Zn1)

coordinated by three cysteine residues and an aspartate. The other $Zn^{2+}$ (Zn2) is coordinated by three cysteine residues from a coiled loop. Zn1 is 5.7 Å apart from Zn2; they are likely connected by an intervening water molecule. **e** Interactions between the core domain and the CTD of TOP3B. **f** Interactions between the CTD of TOP3B and an extended loop of TDRD3. **g** Interactions of TOP3B domain IV with the extended loop of TDRD3 (residues 168−190). Residues contributing to the hydrophobic/hydrophilic interactions are shown as sticks.

stretching forces on domains III and I connecting the two ends of the cleaved G-strand (Fig. 7b). Consequently, domain III moves away from domain I, creating a gap (Fig. 7c) and the topological rearrangement of the DNA produces a swinging rotation of the downstream dsDNA held by the positively charged surfaces of domains II and III. These conformation transitions drive the passage of the T-segment through the domain I-III gate, positioning it among domains I, III, and IV. Under the stretching force, domains III and II oscillate like a spring, and as they return towards domain I (Fig. 7d), the T-segment is trapped by the enzyme (Fig. 7e) and subsequently passed through the second gate (the domain III-IV gate) into the central cavity (Fig. 7f, g). In addition, domain III may disengage temporarily from domain IV as it sways back and forth, facilitating passage of the T-segment. The swinging motion of the downstream dsDNA back to the initial position (Fig. 7g) returns the enzyme to its closed configuration with the T-segment held inside the central cavity. The catalytic center reforms upon gate closure, triggering the rejoining of the G-strand. The whole process introduces +1 unit of DNA linking number. The newly generated DNA wrapping then diffuses into the rest of the DNA substrate via re-opening the topo-gate, consistent with the fact that the gate of a TopoIA was

demonstrated to be flexible enough to trap a DNA segment and subsequently release it[12].

Throughout the DNA relaxation cycle, the G-strand and the dsDNA arm upstream of the cleavage site is topologically constrained by TOP3B. We propose that this constraint is the key to topological transitions of the T-segment and the downstream dsDNA.

## Discussion

Here, we reveal the intermediate states of TOP3B in its DNA and RNA catalytic cycles and an open-gate conformation of TOP3B providing mechanistic insights into the DNA strand-passage mechanism of a Type IA topoisomerase.

We demonstrate that the enzyme is flexible and transitions between open and closed conformations. In our resolved TOP3B dimer complex, the opening of one TOP3B is possibly driven by binding energy from interactions with both the DNA and the second TOP3B. In the context of DNA relaxation, however, the energy needed for gate opening most probably comes from the DNA torsional stress rather than from TOP3B dimerization. The out-of-plane opening mode is expected to remain conserved, as we have also

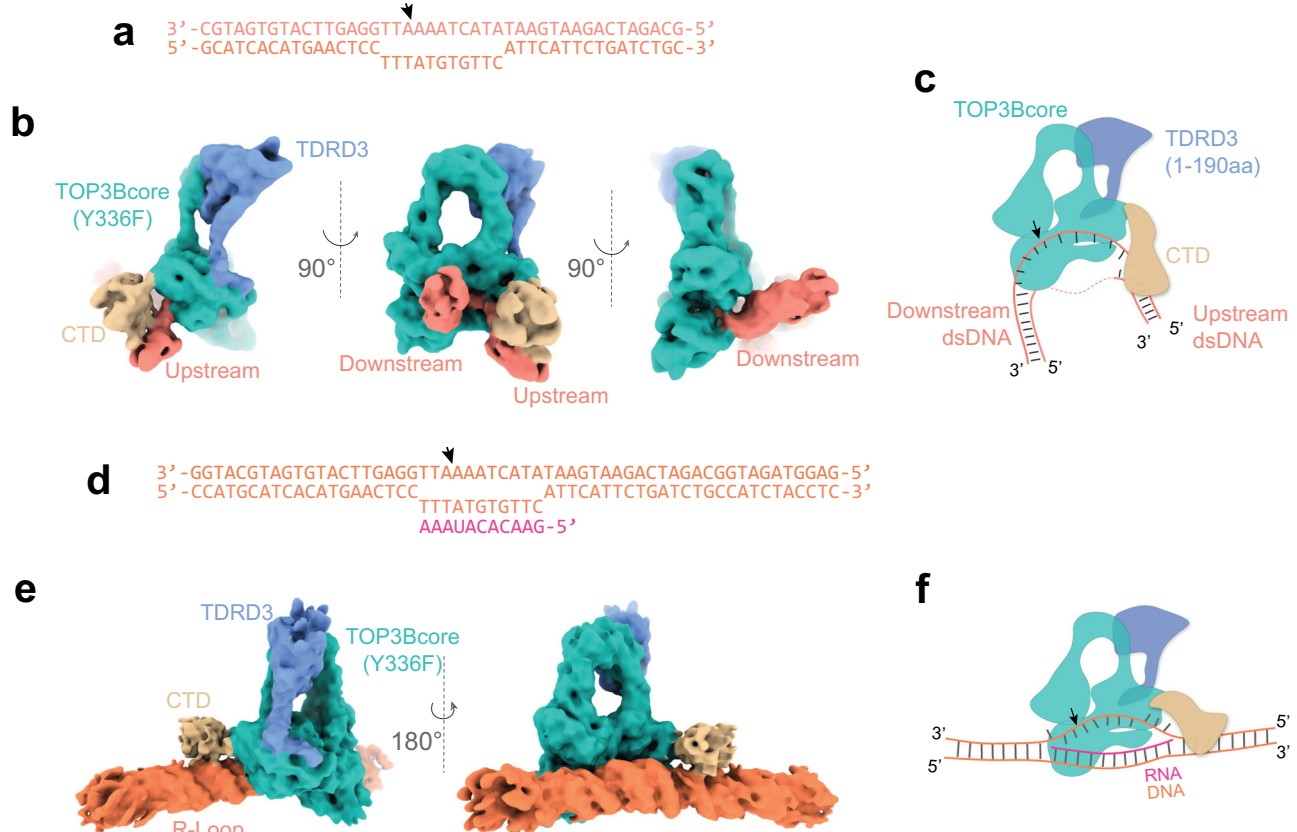

**Fig. 6 | TOP3B-TDRD3 complexes with a DNA bubble and an R-loop. a, b** Cryo-EM volume of TOP3B (full-length) and TDRD3(1-190aa) resolved with a DNA bubble substrate. Additional map density adjacent to TOP3Bcore illustrates the CTD of TOP3B and the dsDNA arm. Arrow indicates the designed TOP3B cleavage site within the bubble region. **c** Schematic representation of the DNA complex of TOP3B-TDRD3 based on the cryo-EM map in (**b**). **d, e** Short R-Loop substrate in complex with TOP3B-TDRD3(1-190aa). **f** Schematic showing the binding of the TOP3B CTD to the dsDNA adjacent to TOP3Bcore.

observed a trend toward this gate-opening within a typical TOP3B catalytic cycle, without TOP3B dimerization (Supplementary Fig. 6f-i).

Opening of the nucleic acid gates between domains I and III and domains III and IV may allow the passage of the T-segment during DNA relaxation by "sequential strand passage". We propose that the gate opening is facilitated by the torsional stress between the two cleaved DNA ends, resulting in the swinging and rotation of the downstream dsDNA. Additionally, in the context of DNA/RNA decatenation, gate opening can be triggered by molecular diffusion of the G-strand relative to the interlocked strand or stimulated by the collision of a translocating helicase or polymerase with the topoisomerase. We report the opening of a TOP3Bcore, which involves a dramatic swinging motion of domains II and III relative to domains I and IV, while the relative orientation of domains II and III remains unchanged. The global movement for this gate opening in TOP3B has not been reported for any TopoIA member previously. A prior structural study on a 30 kD fragment of *E.coli*TOP1 included domains II and III only without I and IV and found some rearrangement between domains II and III, which formed the basis for a proposal of how the TopoIA's DNA gate might open[20]. We also provide new insights on the binding of two divalent cations for TOP3B's DNA/RNA cleavage and rejoining activities. Different DNA backbone and base-stacking configurations can affect the binding affinity of the structural cation $Mn_S^{2+}$, as we could not detect $Mn_S^{2+}$ in the cleavage state when TOP3B bound to a different DNA substrate, but it was consistently present in the rejoining complexes (Supplementary Fig. 4d, f). These two different DNA substrates exhibit different conformations in the TOP3B DNA binding groove (Supplementary Fig. 4b, c). The presence of $Mn_S^{2+}$ in the DNA and RNA

rejoining complexes suggests its essential role in aligning the cleaved DNA ends for re-ligation.

Finally, based on biochemical assays and structural insights, we provide a rationale accounting for the inability of TOP3A to bind RNA to function as an RNA topoisomerase, unlike TOP3B[29].

## Methods

### Preparation of DNA and RNA gap and bubble substrates and TOP3B cleavage assays

DNA/RNA oligos corresponding to the cleavage strands (depicted in the Figs) were Cy3 labeled at the 5' end and annealed with two short oligos to generate the gaped substrates or with a partially complimentary strand to form mismatched bubble substrates.

TOP3B cleavage assays were carried out in buffer comprising 20 mM Bis-Tris (pH 7.0), 100 mM KCl, 0.02% Tween 20, 2 mM DTT, 1 mM MnCl$_2$, with 100 nM DNA/RNA substrate, and 300 nM enzyme or complex. After incubating at 30 °C for 25 min, cleavage products (20 µl) were mixed with 20 µl 2X Formamide gel-loading buffer (10 mM EDTA, 0.025% bromophenol blue, 0.025% Xylene cyanol FF and 0.2% SDS dissolved in formamide), heat-denatured at 95 °C for 2 min, and separated on a 18–20% acrylamide gel containing 7 M Urea, then imaged with a GE Typhoon Phosphorimager. For reversal assays, 400 mM NaCl was introduced to the reaction sample after 25 min incubation and the reaction was stopped by mixing with 2X Formamide gel-loading buffer at indicated time points.

### Cloning, protein expression, and purification

Protein expression vectors encoding hsTOP3Bcore(1-612AA), TOP3B-core-K10M, TOP3Bcore-Y336F, TOP3B-Y336F, hsTDRD3, TDRD3(1-

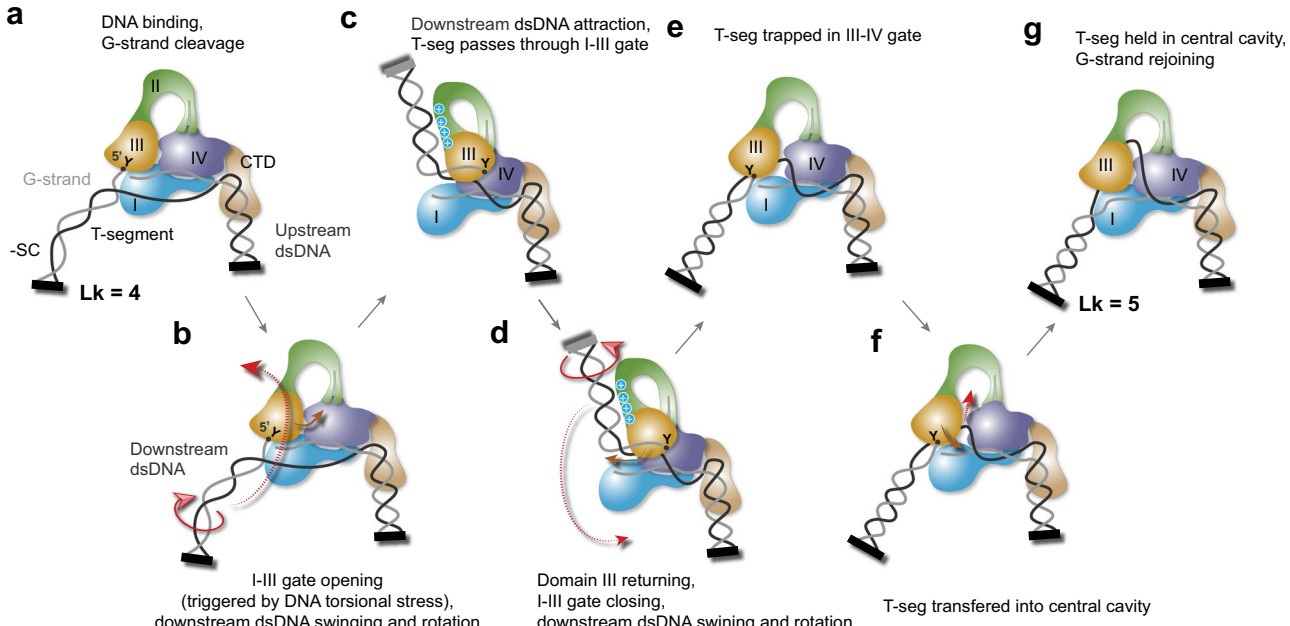

**Fig. 7 | Model of DNA relaxation by TOP3B via a "sequential strand-passage" mechanism. a** TOP3B binds and stabilizes a bubble on negatively supercoiled (-SC) DNA (Linking number = 4) and cleaves the G-strand. Domain III forms a tyrosyl-phosphate bond ("Y") with the 5′-DNA end, while domains I and IV bind non-covalently to the 3′-end. Upstream dsDNA is topologically constrained by the CTD. Black bars indicate topological constraints at the DNA ends. **b**, **c** Tortional stress on DNA ends separates domain III from domain I, coupled with downstream dsDNA rotation and swinging, promoting the passage of the T-segment through the domain I-III gate. Red arrows indicate DNA/domain movements. Downstream dsDNA is temporarily held by the positive charges on domains II and III (blue circles). **d**, **e** Domain III sways back as the domain I-III gate closes. Domain III-IV gate temporarily opens during this conformational transition and traps the T-segment. The supercoiled downstream dsDNA continues rotating, swinging back to the original position, facilitating the passage of the T-segment into the central cavity. This process introduced +1 unit of DNA linking number and dissipates the DNA negative supercoiling. **f**, **g** The enzyme resets to the closed state with the T-segment held in the central cavity.

171AA), and TDRD3(1-190AA) were individually inserted into a pLEXm plasmid featuring an N-terminal His8-MBP-tag[10]. The resulting vectors containing the recombinant DNA were then transfected into human HEK293T cells to initiate protein expression. The detailed methods for vector transfection, cell harvesting, cell lysis, as well as the subsequent purification steps involving affinity column and SEC column purification were previously documented[10,55], except that the N-terminal His8-MBP-tags of all protein components were removed by PreScission Protease (cytiva) prior to loading onto the SEC column.

### Cryo-EM sample preparation and vitrification
DNA and RNA pre-cleavage state complexes of TOP3B were prepared by mixing cTOP3B-Y336F or TOP3B-Y336F, complexed with full-length or truncated TDRD3, with equimolar amounts of the designated gaped DNA/RNA substrate in a buffer comprising 25 mM Bis-Tris (pH 6.8), 0.1 M KCl, 0.5 mM MnCl₂, 0.1 % Octyl β-D-glucopyranoside, 4 mM DTT and 2% Glycerol, incubated at RT for 20 min before proceeding to grid preparation. The concentration of the heterodimer was maintained at 3 mg/ml. Post-cleavage DNA complexes were prepared with cTOP3B and equimolar amount of gaped DNA substrate using identical reaction conditions as above. DNA and RNA rejoining complexes were prepared with cTOP3B-K10M, mixed with the gaped DNA/RNA substrate in a buffer resembling the above except for a higher, 5 mM, MnCl₂ concentration. The mixtures were then incubated at RT for 2 h before advancing to grid preparation. During the grid preparation process, 3 µl of each Cryo-EM sample was applied onto the plasma-treated (PELCO easiGlow) TEM grids (Quantifoil 300 mech Copper/Gold, R1.2/1.3) inside the climate chamber of the Vitrobot Mark IV (FEI). After waiting for 15 s at 4 °C and 100% humidity, the grids bearing the sample were blotted with filter paper for 2.5 s and plunge-frozen in liquid ethane to achieve sample vitrification.

### Cryo-EM data collection and image processing
TEM Grids with samples were imaged on a Titan Krios microscope (Thermo Fisher) at 300 kV with a K2 Summit direct electron detector (Gatan), or on an Talos Arctica microscope (Thermo Fisher) at 200 kV with K3 detector (Gatan). Both microscopes contained a Gatan Bio-quatum energy Filter. Movies were recorded in counting mode with a super resolution pixel size of 0.415 Å (0.83 Å physical pixel size), with a defocus range of 0.8–2.4 µM. Each raw movie stack has a total dose of ~40–50 e-/Å² and 40–50 frames, with an exposure time of ~2.5 s.

Raw movies were processed with cryoSPARC (v3.2 or v4.2). They were first motion corrected and binned to 0.83 Å/pixel with the Patch Motion Correction job, followed by CTF estimation using Patch CTF. Motion-corrected micrographs with resolution better than 6.5 Å were selected for subsequent particle-picking step using Blob Picker and Inspect Picks tools. After two rounds of 2D classification, good particles representing various 2D projections of the protein complexes were selected for the Ab-Initio job to generate the 3D models, which were further refined with Homogeneous Refinement jobs and finally processed with Local Refinement job using a soft mask generated with the Volume Tools job (Supplementary Fig. 8). The resulting cryo-EM volumes were sharpened using the DeepEMhancer package.

### Model building
For building the DNA and RNA complexes of the TOP3Bcore-OB, the crystal structure of the apo hsTOP3Bcore-OB (PDB 5gve) was used as template, rigidly fitted into the cryo-EM volumes of the DNA and RNA complexes of TOP3Bcore-OB in UCSF Chimera and refined in Coot. Intact and cleaved ssDNA/RNA substrate observed in the maps was placed accordingly in Coot. The side chains of the mutated amino acid residues in the Y336F and K10M mutants were replaced. For the apo structure of the full-length TOP3B complexed with TDRD3, the crystal structure of hsTOP3Bcore-OB (PDB 5gve) and the AlphaFold predicted

C-terminal Domains of TOP3B were used as templates. The models were then iteratively refined in Phenix and Coot (Supplementary Fig. 9).

## Reporting summary

Further information on research design is available in the Nature Portfolio Reporting Summary linked to this article.

## Data availability

The cryo-EM density maps and corresponding atomic coordinates were respectively deposited in the EMDB and PDB: EMD-45376 and 9C9Y for the cTOP3B DNA pre-cleavage complex, EMD-45378 and 9CA0 for the cTOP3B DNA post-cleavage complex, EMD-45379 and 9CA1 for the cTOP3B DNA rejoining complex, EMD-45390 and 9CAG for the cTOP3B RNA pre-cleavage complex, EMD-45380 and 9CA4 for the cTOP3B RNA rejoining complex, EMD-45374 and 9C9W for the cTOP3B dimer with DNA bubble, EMD-45391 and 9CAH for the Full-length TOP3B-TDRD3 apo complex. For the cTOP3B DNA pre-cleavage, post-cleavage and rejoining complexes with the second DNA gap substrate (presented in Supplementary Fig. 4), the map/model access codes are: EMD-45393/9CAJ, EMD-45395/9CAL and EMD-45394/9CAK. Access codes for other published atomic coordinates used for comparison purposes were provided accordingly in the manuscript and figure legends, including: 5GVC, 5GVE, 4CGY, 1D6M, 6CQ2 and 1I7D. Source data are provided with this paper.

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

## Acknowledgements

Cryo-EM data were collected at the Center for Structural Biology cryo-EM facility (NCI Frederick), NICE-NIH Intramural Cryo-EM Consortium, NIDDK, and the MICEF Cryo-EM Facility. We thank Dan Shi, Mi Li, Rick Huang, Yanxiang Cui, Huaibin Wang, AJ Morton, Zabrina Lang, and Ulrich Baxa for assistance with cryo-EM data collection. We thank Keir C Neuman (NHLBI) for his diligent proofreading of this manuscript and valuable insights provided on the TOP3B model. This work was supported by the Center for Cancer Research, the Intramural Program of the National Cancer Institute, NIH (Z01-BC006161) to Y.P. and NIDDK (DK075037) to W.Y.

## Author contributions

X.Y., W.Y., and Y.P. designed experiments and prepared the manuscript. W.Y. and Y.P. supervised the study. X.Y. performed the biochemical assays, prepared cryo-EM samples, collected and processed cryo-EM data, and built the protein atomic models. X.C. and W.Y. helped on model building. X.Y. and W.Y. performed structural analysis.

## Funding

## Competing interests

The authors declare no competing interests.
