## [Peer Review file · Nature Communications]

Structural insights into human Topoisomerase 3 β DNA and RNA catalysis and nucleic acid gate dynamics

Corresponding Author: Dr Xi Yang

Version 0:

Reviewer comments:

Reviewer #1

(Remarks to the Author)

In this article, the authors address a long-standing knowledge gap regarding the specific RNA/DNA cleavage and rejoining mechanisms of human TOP3B. They solved a series of high-resolution Cryo-EM structures that capture the pre-, post-, and transitional states of TOP3B, providing novel insights into its function. Based on these structural data, the authors propose mechanisms for DNA cleavage, end joining, and a sequential strand-passage mechanism during DNA relaxation. Additionally, the comparison of these structures with those of TOP3A provides valuable insights into the dual RNA/DNA functionality of TOP3B. According to the PDB validation reports, the models are well constructed. All methods are appropriate, meet the expected standards in the field, and the manuscript provides sufficient detail to allow others to reproduce the work.

This work represents a structural tour de force, shedding light on several debated aspects of TOP3B's mechanism, such as the role of metal ions, which the authors reveal to have a unique release and recapture mechanism coupled with large conformational changes. The data presented in this paper are compelling and strongly support the proposed mechanisms for TOP3B function. The manuscript is exceptionally clear and well-written, with particularly effective figures. This reviewer appreciates the informative and well-organized extended data, which directly supports the main findings. The mechanistic figures, in particular, are highly effective. The authors' decision to discuss and compare their data to other systems within the main body of the manuscript was appropriate, and they clearly and concisely walk the reader through a complex story. Overall, this manuscript was a pleasure to review, and I have only a few minor suggestions that the authors may find beneficial:

1. Lines 63-73: In the sections discussing mutations that inactivate catalytic activity or rejoining, it would be helpful to provide more detail on the rationale for selecting specific mutations. For example, "Y336's aromatic hydroxyl group functions as the attacking nucleophile promoting nucleic acid cleavage. To capture a pre-catalytic structure, a Y->F mutation was used to preserve potential stacking interactions while preventing strand cleavage."
2. Lines 82 and 104: The choice between Mn and Mg ions was not clearly explained. Can these metals easily substitute for one another, or do certain enzymes exhibit a strong preference for one? The authors might direct readers to their biochemical cleavage assays in the extended data to show that the complex remains catalytically active under both conditions.
3. Lines 221-222: The authors speculate that the enhanced rejoining activity for RNA may depend on the 2'-OH group adjacent to the scissile phosphate. Could the authors test this hypothesis by comparing the activity of 2'-OH and 2'-OMe substrates in their biochemical assays? This would be a relatively simple and quick experiment to provide further insight into this fascinating aspect of the manuscript.
4. Dimer structure: The presence of a dimer is interesting and warrants further discussion. Is this purely artifactual, or could there be biological relevance? Does the dimer interaction artificially influence the open-gate state?
5. Sequential strand-passage mechanism: The authors do an excellent job of guiding readers through the complex nucleic acid remodeling steps involved. However, slight adjustments to Figure 7 could further enhance clarity:

- a) Increase the prominence of the red arrows indicating twists and large-scale movements, and refer to these more explicitly in the text to help guide readers through the mechanism.
- b) Increase the font size of the starting and ending linking numbers, as they are important for illustrating the overall effect but are currently difficult to read.

Reviewer #3

(Remarks to the Author)

Yang et al. describe structural characterization of the mechanism of human Topoisomerase 3beta. The manuscript presents numerous cryo-EM structures of the enzyme in different states. Overall, this is high-quality and technically solid work which provides valuable insights into the mechanism of TOPO3B, an enzyme important for ensuring genome integrity.

The first part of the manuscript describes a series of structures corresponding to the different steps of catalysis. Structures corresponding to pre-cleavage, post-cleavage and rejoining states are presented. Each step is obtained using appropriate mutant proteins. These structures elegantly show the configuration of the active site at each step with one catalytic and one structural metal ion, and changes in the alignment of key reactants at the active site. Both Mg²⁺ and Mn²⁺ support the enzymatic reaction (PMID: 35945419). The authors need to comment on which of these is the physiological cofactor and why Mn²⁺ is a good choice for structural studies.

TOP3B can process both DNA and RNA, and structures in complex with RNA are also presented. RNA binding is similar to DNA binding but requires conformational changes of the entire protein structure. The authors hypothesize that this ability is unique to TOPO3B.

Authors also used a bubble DNA and determined the structure in which two TOPO3B molecules are bound to this DNA. One of the molecules has an altered conformation where the ring of TOPO3B is open. The authors interpret this state as an open gate conformation that allows one strand to pass through. I have a few comments/questions here. What exactly is the observed dimer state? Is this an artifact of sample preparation or may it have some physiological relevance? Are there any protein-protein contacts in the dimer? Are the residues involved in these contacts conserved? Can a full-length enzyme containing the CTD adopt such a dimer configuration? Is it possible that the presence of the other protein copy (the closed conformation) in this complex can influence the conformation of the open form?

The authors mention crystal structures of a fragment of E. coli TOP1 in different conformations. Are the conformational changes observed in these structures similar to those observed between the closed and open structures of TOPO3B? Comparison with these structures could strengthen the interpretation of the observed conformational changes of TOPO3B.

Version 1:

Reviewer comments:

Reviewer #1

(Remarks to the Author)

The original submission was very high quality, with both initial reviewers expressing strong support and recommending only minor revisions. The authors have thoughtfully addressed each point raised, providing clear and satisfactory responses that improve the manuscript's clarity and impact. I find no further revisions necessary and fully endorse this manuscript for its scientific merit and contribution to the field. The study's conclusions are well-supported, and the scope and significance align with the high standards of quality and impact characteristic of Nature Communications.

Reviewer #3

(Remarks to the Author)

The authors have addressed my comments appropriately and the paper should be accepted.

Point-by-point response to reviewer comments

Response to Reviewer #1:

In this article, the authors address a long-standing knowledge gap regarding the specific RNA/DNA cleavage and rejoining mechanisms of human TOP3B. They solved a series of high-resolution Cryo-EM structures that capture the pre-, post-, and transitional states of TOP3B, providing novel insights into its function. Based on these structural data, the authors propose mechanisms for DNA cleavage, end joining, and a sequential strand-passage mechanism during DNA relaxation. Additionally, the comparison of these structures with those of TOP3A provides valuable insights into the dual RNA/DNA functionality of TOP3B. According to the PDB validation reports, the models are well constructed. All methods are appropriate, meet the expected standards in the field, and the manuscript provides sufficient detail to allow others to reproduce the work.

This work represents a structural tour de force, shedding light on several debated aspects of TOP3B's mechanism, such as the role of metal ions, which the authors reveal to have a unique release and recapture mechanism coupled with large conformational changes. The data presented in this paper are compelling and strongly support the proposed mechanisms for TOP3B function. The manuscript is exceptionally clear and well-written, with particularly effective figures. This reviewer appreciates the informative and well-organized extended data, which directly supports the main findings. The mechanistic figures, in particular, are highly effective. The authors' decision to discuss and compare their data to other systems within the main body of the manuscript was appropriate, and they clearly and concisely walk the reader through a complex story. Overall, this manuscript was a pleasure to review, and I have only a few minor suggestions that the authors may find beneficial:

We sincerely thank the reviewer for his/her thoughtful and encouraging comments on our work.

1. Lines 63-73: In the sections discussing mutations that inactivate catalytic activity or rejoining, it would be helpful to provide more detail on the rationale for selecting specific mutations. For example, "Y336's aromatic hydroxyl group functions as the attacking nucleophile promoting nucleic acid cleavage. To capture a pre-catalytic structure, a Y->F mutation was used to preserve potential stacking interactions while preventing strand cleavage."

Thank you for the suggestion. We have added further details in the manuscript to clarify the rationale behind selecting the Y336F and K10M mutations:

“We substituted the nucleophile Y336, which forms tyrosyl-phosphate covalent bond with cleaved DNA, with a phenylalanine (F), to disable nucleic acid cleavage while preserving the DNA/RNA interactions.”

“We obtained a DNA post-cleavage complex with wild-type cTOP3B, and generated DNA and RNA rejoining complexes (Fig. 1f) with the rejoining-deficient mutant cTOP3B-K10M (detailed in the corresponding section).”

2. Lines 82 and 104: The choice between Mn and Mg ions was not clearly explained. Can these metals easily substitute for one another, or do certain enzymes exhibit a strong preference for one? The authors might direct readers to their biochemical cleavage assays in the extended data to show that the complex remains catalytically active under both conditions.

We appreciate this valuable suggestion. We have clarified our rationale in the revised manuscript for choosing Mn^{2+} over Mg^{2+} and included the following statement: “ Mg^{2+} and Mn^{2+} both activate TOP3B in nucleic acid cleavage and rejoining, with Mn^{2+} being more effective¹⁰ owing to its less stringent coordination requirement and better ability to stabilize certain reaction intermediates. Additionally, the higher electron count of Mn^{2+} can enhance visibility in cryo-EM maps, making it the preferred choice for our study.”

3. Lines 221-222: The authors speculate that the enhanced rejoining activity for RNA may depend on the 2'-OH group adjacent to the scissile phosphate. Could the authors test this hypothesis by comparing the activity of 2'-OH and 2'-OMe substrates in their biochemical assays? This would be a relatively simple and quick experiment to provide further insight into this fascinating aspect of the manuscript.

Thanks for the suggestion. We agree that testing 2'-OMe may be a good way to investigate the role of the 2'-OH group in RNA rejoining. Additionally, insights from well-characterized catalytic processes similar to TOP3B rejoining, such as nucleotide incorporation by polymerases, may also be informative. For instance, human DNA polymerase η (S113A mutant) incorporates nucleotides more efficiently with a ribonucleotide at the primer terminus. The enzyme uses the 2'-OH to facilitate deprotonation the 3'-OH as the nucleophile (doi.org/10.1073/pnas.2103990118, Fig. 4). In addition to a reactive 2'-OH group, a ribonucleotide may also dictate a different sugar pucker (3'-endo) than a deoxyribonucleotide (2'- or 3'-endo), which could further increase its reactivity. But the bulky nature of the methyl group of 2'-OMe may interfere with TOP3B. In addition, the methyl group cannot serve as proton shuttle. Thus, we are not sure if 2'-OMe can verify our hypothesis.

4. Dimer structure: The presence of a dimer is interesting and warrants further discussion. Is this purely artifactual, or could there be biological relevance? Does the dimer interaction artificially influence the open-gate state?

Thank you for bringing up this point. The dimer complex observed in our cryo-EM data represents a minor particle population observed when the sample was prepared at relatively high protein concentrations (3 mg/ml), suggesting that it may be an artifact not typically present at physiological protein concentrations in cells. However, we cannot entirely rule out a functional role for TOP3B dimers in specific DNA/RNA transactions.

We suspect that the opening of one TOP3B molecule is driven by binding energy from interactions with both the DNA and the second TOP3B, upon dimerization. In the context of DNA relaxation, the energy required to open the protein gate should derive from the torsional stress of DNA, instead of from the protein dimerization. However, the out-of-plane opening mode may remain conserved in both scenarios, as it may represent a low-energy conformational transition. Even in the absence of dimerization, we have observed a trend of this gate opening within a typical TOP3B catalytic cycle (Extended Data Fig. 6f-i). We have added the above in the Discussion section.

5. Sequential strand-passage mechanism: The authors do an excellent job of guiding readers through the complex nucleic acid remodeling steps involved. However, slight adjustments to Figure 7 could further enhance clarity:

a) Increase the prominence of the red arrows indicating twists and large-scale movements, and refer to these more explicitly in the text to help guide readers through the mechanism.

b) Increase the font size of the starting and ending linking numbers, as they are important for illustrating the overall effect but are currently difficult to read.

Thank you for these suggestions. We have modified the figure accordingly.

Response to Reviewer #3:

Yang et al. describe structural characterization of the mechanism of human Topoisomerase 3beta. The manuscript presents numerous cryo-EM structures of the enzyme in different states. Overall,

this is high-quality and technically solid work which provides valuable insights into the mechanism of TOPO3B, an enzyme important for ensuring genome integrity.

The first part of the manuscript describes a series of structures corresponding to the different steps of catalysis. Structures corresponding to pre-cleavage, post-cleavage and rejoining states are presented. Each step is obtained using appropriate mutant proteins. These structures elegantly show the configuration of the active site at each step with one catalytic and one structural metal ion, and changes in the alignment of key reactants at the active site.

We appreciate the reviewer's positive feedback.

Both Mg²⁺ and Mn²⁺ support the enzymatic reaction (PMID: 35945419). The authors need to comment on which of these is the physiological cofactor and why Mn²⁺ is a good choice for structural studies.

Thank you for the suggestion. We have included our rationale for choosing Mn²⁺ over Mg²⁺ in the revised manuscript as follows: "Mg²⁺ and Mn²⁺ both activate TOP3B in nucleic acid cleavage and rejoining, with Mn²⁺ being more effective¹⁰ owing to its less stringent coordination requirement and better ability to stabilize certain reaction intermediates. Additionally, the higher electron count of Mn²⁺ can enhance visibility in cryo-EM maps, making it the preferred choice for our study."

TOP3B can process both DNA and RNA, and structures in complex with RNA are also presented. RNA binding is similar to DNA binding but requires conformational changes of the entire protein structure. The authors hypothesize that this ability is unique to TOPO3B. Authors also used a bubble DNA and determined the structure in which two TOPO3B molecules are bound to this DNA. One of the molecules has an altered conformation where the ring of TOPO3B is open. The authors interpret this state as an open gate conformation that allows one strand to pass through. I have a few comments/questions here. What exactly is the observed dimer state? Is this an artifact of sample preparation or may it have some physiological relevance?

Thank you for the insightful questions on the TOP3B dimer structure. The dimer complex represents a minor particle population observed in our cryo-EM samples. It is likely an artifact resulting from the high protein concentration (3 mg/ml) used during cryo-EM preparation, not representing a functional state of TOP3B at physiological protein concentrations in cells. But we do not have evidence that the gate opening during DNA relaxation depends on dimerization of TOP3B. Please see our response to Review #1, and the added discussion in our revised manuscript.

Are there any protein-protein contacts in the dimer? Are the residues involved in these contacts conserved?

We identified interactions at the dimeric interface of TOP3B (Extended Data Fig. 5), which may stabilize the TOP3B dimer-DNA complex. However, the amino acid pairs mediating these protein-protein interactions—D365-K394, Y362-S335, Y329-N347, and N347-R407—are not conserved in either TOP3A or *E. coli*TOP3.

Can a full-length enzyme containing the CTD adopt such a dimer configuration?

Dimerization is not required and unlikely involved in real DNA relaxation. We propose that the proper binding of full-length TOP3B to a DNA bubble (typically on negatively supercoiled DNA) initiates cleavage of one DNA strand and triggers rotation of the downstream dsDNA, which then promotes the gap opening and transfer of the other continuous single strand into the enzyme's central cavity (Fig. 7 and Fig. 4g,h). This minimizes the likelihood of two TOP3B molecules binding to a single DNA bubble.

Is it possible that the presence of the other protein copy (the closed conformation) in this complex can influence the conformation of the open form?

Yes, it seems that in the dimeric complex, the opening of one TOP3B is driven by binding energy from interactions with both the DNA and the second TOP3B. In the context of DNA relaxation, however, the energy needed to open the protein gate comes from the torsional stress of DNA rather than from TOP3B dimerization. This out-of-plane opening mode is expected to remain conserved because in the absence of TOP3B dimerization, we observe a trend toward this gate-opening within a typical TOP3B catalytic cycle (Extended Data Fig. 6f-i). We have added the above discussion in the Discussion section.

The authors mention crystal structures of a fragment of *E. coli* TOP1 in different conformations. Are the conformational changes observed in these structures similar to those observed between the closed and open structures of TOPO3B? Comparison with these structures could strengthen the interpretation of the observed conformational changes of TOPO3B.

Thank you for the suggestion to compare TOP3B with the previously published conformations of the *E. coli*TOP1 protein fragments. The out-of-plane opening of the TOP3B gate has not been observed before and reveals a significant swinging motion of domains II and III relative to domains I and IV. In contrast, the published *E. coli*TOP1 structures (PDB 1CY9 and 1CYY) contain domains II and III only. Without domains I and IV, we can only compare domains II and III and cannot compare the global conformational changes involved in gate opening. Additionally, the *E. coli*TOP1 structures exhibited a domain

rearrangement between II and III, which we did not observe in TOP3B. We have added this comparison to the Discussion section.

Point-by-point response to reviewer comments

Response to Reviewer #1:

Reviewer #1 (Remarks to the Author):

The original submission was very high quality, with both initial reviewers expressing strong support and recommending only minor revisions. The authors have thoughtfully addressed each point raised, providing clear and satisfactory responses that improve the manuscript's clarity and impact. I find no further revisions necessary and fully endorse this manuscript for its scientific merit and contribution to the field. The study's conclusions are well-supported, and the scope and significance align with the high standards of quality and impact characteristic of Nature Communications.

We sincerely appreciate the reviewer's support and thoughtful suggestions which have significantly enhanced the clarity and quality of the manuscript.

Response to Reviewer #3:

Reviewer #3 (Remarks to the Author):

The authors have addressed my comments appropriately and the paper should be accepted.

We are pleased that our revisions have addressed the reviewer's comments and questions. We sincerely appreciate the reviewer's thoughtful suggestions which have significantly enhanced the clarity and quality of the manuscript.